# ERRα Up-Regulates Invadopodia Formation by Targeting HMGCS1 to Promote Endometrial Cancer Invasion and Metastasis

**DOI:** 10.3390/ijms24044010

**Published:** 2023-02-16

**Authors:** Shuting Tang, Jincheng Ma, Pingping Su, Huifang Lei, Yao Tong, Liangzhi Cai, Shuxia Xu, Xiaodan Mao, Pengming Sun

**Affiliations:** 1Laboratory of Gynecologic Oncology, Fujian Maternity and Child Health Hospital, College of Clinical Medicine for Obstetrics & Gynecology and Pediatrics, Fujian Medical University, Fuzhou 350001, China; 2Department of Gynecology, Fujian Provincial Maternity and Child Hospital, College of Clinical Medicine for Obstetrics & Gynecology and Pediatrics, Fujian Medical University, Fuzhou 350001, China; 3Department of Pathology, Fujian Maternity and Child Health Hospital, College of Clinical Medicine for Obstetrics & Gynecology and Pediatrics, Fujian Medical University, Fuzhou 350001, China; 4Fujian Key Laboratory of Women and Children’s Critical Diseases Research, Fujian Maternity and Child Health Hospital (Fujian Women and Children’s Hospital), Fuzhou 350001, China; 5Fujian Clinical Research Center for Gynecologial Oncology, Fujian Maternity and Child Health Hospital (Fujian Obstetrics and Gynecology Hospital), Fuzhou 350001, China

**Keywords:** endometrial cancer, ERRα, HMGCS1, intracellular cholesterol metabolism, invadopodia, epithelial–mesenchymal transition pathway, metastasis

## Abstract

Estrogen-related receptor alpha (ERRα) plays an important role in endometrial cancer (EC) progression. However, the biological roles of ERRα in EC invasion and metastasis are not clear. This study aimed to investigate the role of ERRα and 3-hydroxy-3-methylglutaryl-CoA synthase 1 (HMGCS1) in regulating intracellular cholesterol metabolism to promote EC progression. ERRα and HMGCS1 interactions were detected by co-immunoprecipitation, and the effects of ERRα/HMGCS1 on the metastasis of EC were investigated by wound-healing and transwell chamber invasion assays. Cellular cholesterol content was measured to verify the relationship between ERRα and cellular cholesterol metabolism. Additionally, immunohistochemistry was performed to confirm that ERRα and HMGCS1 were related to EC progression. Furthermore, the mechanism was investigated using loss-of-function and gain-of-function assays or treatment with simvastatin. High expression levels of ERRα and HMGCS1 promoted intracellular cholesterol metabolism for invadopodia formation. Moreover, inhibiting ERRα and HMGCS1 expression significantly weakened the malignant progression of EC in vitro and in vivo. Our functional analysis showed that ERRα promoted EC invasion and metastasis through the HMGCS1-mediated intracellular cholesterol metabolism pathway, which was dependent on the epithelial–mesenchymal transition pathway. Our findings suggest that ERRα and HMGCS1 are potential targets to suppress EC progression.

## 1. Introduction

Endometrial cancer (EC) is currently the second most common gynecological tumor worldwide and one of the most common gynecological malignancies in developed countries [1,2]. In 2021, there were approximately 66,570 new cases of EC and 12,940 reported deaths resulting from EC in the United States [3]. According to the National Cancer Center, in 2019, the incidence rate of EC was 10.28 per 100,000 and the mortality rate was 1.9 per 100,000 in China [2]. Both the incidence and associated mortality of EC have been increasing recently [4]. Patients with advanced EC have a relatively low 5-year relative survival rate and unfavorable prognosis, primarily due to invasion and metastasis to distant target organs [5]. Currently, women with recurrent, advanced, or metastatic disease often have poor survival outcomes without therapeutic options. However, some studies have provided new perspectives on immunotherapy for these diseases, which may improve the outcomes of patients with gynecological cancers [6]. A recent study has provided an overview of the development of clinical immunotherapy for advanced or recurrent EC, and offered a new strategy for EC treatment [7]. As a risk factor associated with EC, obesity also plays a vital role in the development of EC. An increase in the incidence of obesity may result in an increased incidence of EC. Meanwhile, abnormal cholesterol metabolism has been reported in obese people [8]. Recently, metformin has been shown to reverse the invasion of advanced EC in vivo. A previous study using a transgenic mouse model of EC indicated that the levels of lipolysis and lipid peroxidation are significantly higher in obese mice than in lean mice. It was further shown that metformin effectively down-regulates the rate of lipolysis and peroxidation of related lipids. The same study suggested that obesity is an unfavorable factor for EC invasion and metastasis, and also that the invasion of EC is associated with elevated cholesterol levels in the tumor cells [9]. Recently, cholesterol metabolism has become an intriguing topic in oncology research. However, the specific mechanism responsible for abnormal cholesterol metabolism during the progression of EC remains unclear [10].

Estrogen-related receptor alpha (ERRα) belongs to a nuclear receptor superfamily. It can activate the transcription of genes downstream of estrogen response elements in the absence of natural ligands [11]. It is also considered to be a principal regulator of tumor energy metabolism [12,13]. Recently, ERRα has been confirmed as a marker of poor prognosis in breast cancer, prostate cancer [14], and colorectal cancer [15]. Similarly, our team found that patients with EC and high ERRα expression levels have a higher risk of deep myometrial invasion [16], which suggests that ERRα is a potential therapeutic target for EC. Meanwhile, 3-hydroxy-3-methyl-glutaryl-coenzyme A reductase (HMGCR) is known as the rate-limiting enzyme of the mevalonate pathway. Besides HMGCR, other enzymes have also been shown to serve as control points for cholesterol synthesis [17]. 3-Hydroxy-3-methylglutaryl-CoA synthase 1 (HMGCS1) is a cytoplasmic enzyme that functions upstream of HMGCR, and there is evidence that HMGCS1 is one of the potential regulatory points in the mevalonate pathway [18]. It has been reported that HMGCS1 is positively associated with the proliferation, migration, and invasion of gastric cancer cells [19]. In hepatocellular carcinoma, abnormal expression levels of cholesterol-metabolism-related genes, such as sterol regulatory element-binding protein 2, HMGCS1, and HMGCR, leads to the lipid reprogramming of tumor cells and promotes the formation of invasive pseudopods [20]. These findings indicate that HMGCS1 is associated with tumor invasion and metastasis and is likely to be a new tumor treatment target. 

In recent years, many findings have suggested that abnormal lipid and cholesterol metabolism are involved in the malignant progression of tumors [21,22]. Cholesterol is an essential component of mammalian cell membranes and it significantly affects membrane fluidity, permeability, and curvature, and the interactions of membrane proteins [23]. Moreover, cholesterol together with other lipids, form important structures in cell membranes that alter signal transduction, thus affecting cell proliferation and migration [24]. Invadopodia, other characteristic membrane structures in cancer cells that can induce migration and invasion, are closely regulated by lipid and cholesterol metabolism. With the help of invadopodia, cancer cells degrade the extracellular matrix, eventually leading to distant metastasis [25]. However, the mechanism whereby cholesterol metabolism in the cell membrane affects the progression of EC remains to be investigated. In addition, epithelial–mesenchymal transition (EMT) can make polarized epithelial cells invasive and metastatic [26]. We propose that EMT may also be vital to the invasion and metastasis of EC, as EC is a common epithelial malignancy.

The role of ERRα and HMGCS1 in cholesterol metabolism and how this mechanism contributes to invadopodia formation and tumor metastasis require further investigation. We hypothesize that ERRα regulates cancer cell membrane structure and function through HMGCS1-mediated intracellular cholesterol metabolism to promote tumor invasion and metastasis. This may provide the basis for ERRα and HMGCS1 clinical application in precision diagnosis and treatment of metastatic EC.

## 2. Results

### 2.1. Serum Lipid Levels in Patients with EC Are Associated with the Progression of EC

An analysis of serum samples showed that patients with EC had higher triglyceride (TG) levels, but lower apolipoprotein A (APO-A) and high-density lipoprotein (HDL) than the healthy population (*p* < 0.05, Figure 1a). Meanwhile, compared to patients with non-endometrioid endometrial cancer (NEEC), those with endometrioid endometrial cancer(EEC) had higher low-density lipoprotein (LDL) levels (*p* < 0.05, Figure 1b). We further found that patients with advanced-stage EC had higher LDL levels than those with early-stage EC (*p* < 0.05, Figure 1c). Moreover, patients with lymph node metastasis (LNM) had higher LDL levels than those without LNM (*p* < 0.05, Figure 1d). The TG concentration was higher in patients with deep myometrial invasion (DMI) than those with shallow myometrial invasion (SMI), but there was no statistically significant difference (*p* < 0.05, Figure 1e). Altogether, these clinical data suggested that serum lipid levels in patients with EC are closely associated with EC progression.

### 2.2. ERRα and HMGCS1 Are Highly Expressed in the Tissues of Patients with EC and Are Associated with Metastasis

To analyze the levels of ERRα and HMGCS1 in EC, immunohistochemical (IHC) assays were performed on a tissue microarray (TMA) comprising 73 EC tissues and 30 control endometrial tissues (Table 1). ERRα immunoreactivity was significantly higher in EC tissues than control tissues (*p* < 0.001, Figure 2a & Table 2). HMGCS1 was also detected in 103 tissue specimens, and its immunoreactivity was higher in 73 cancer tissues than in 30 normal endometrial tissues (*p* < 0.001, Figure 2a & Table 2). Furthermore, a positive correlation existed between HMGCS1 and ERRα immunoreactivity, based on Pearson’s rank correlation analysis (r = 0.39, *p* < 0.05, Figure 2b). The clinical correlation of the targeted predictors was also analyzed. The results showed that the high expression levels of ERRα, HMGCS1 and ERRα/HMGCS1 were not different between different tumor stages, histological grades, pathological types, and LNM status (*p* > 0.05, Figure 2c & Table 2). However, the high expression levels of ERRα, HMGCS1 and ERRα/HMGCS1 were distinct between the SMI (<1/2 MI) and DMI (≥1/2 MI) groups (*p* < 0.05, Figure 2c & Table 2). 

Intriguingly, a study of serum lipids from the same population showed that the levels of TG, LDL, and APO-B were significantly higher, while the levels of HDL and APO-A were lower in patients with EC than in control individuals (*p* < 0.05, Figure 2d & Table 3). These results indicated that ERRα and HMGCS1 are predictors of poor prognosis for EC patients. Furthermore, the increase in ERRα and HMGCS1 levels were involved in the invasion and metastasis of EC.

### 2.3. Simvastatin Inhibits Epithelial–Mesenchymal Transition in EC

Simvastatin, a 3-hydroxy-3-methylglutaryl coenzyme A reductase inhibitor, shows potent anti-tumor efficacy against EC [27]. We came to the same conclusion. As shown in Figure 3a, compared to the controls, the scratched spaces decreased by 57.14% in KLE cells and 81.87% in HEC-1A cells treated with simvastatin for 24 h (*p* < 0.05, Figure 3a,b). A similar trend was noted in transwell assays, in which the invasive ability of KLE and HEC-1A cells was decreased by 62.44% and 50.54% (*p* < 0.05, Figure 4c,d), respectively, after simvastatin treatment. Furthermore, to establish that simvastatin had an inhibitory effect on HMGCS1, we examined HMGCS1 levels after treatment with simvastatin for 24 h. There was a significant decrease in HMGCS1 expression levels in both EC cell lines (*p* < 0.05, Figure 3e,f). Moreover, to explore whether inhibiting the expression of HMGCS1 reduces the accumulation of other adhesion molecules, thereby inhibiting EMT, we detected changes in the expression levels of EMT-associated signaling molecules after simvastatin treatment. Simvastatin treatment decreased the expression level of HMGCS1 and vimentin, but significantly up-regulated the expression of E-cadherin (E-cad), an epithelial phenotype protein, in EC cells (*p* <0.05, Figure 3g,h). These results indicated that simvastatin prevents EC cells from metastasizing and invading by inhibiting EMT.

### 2.4. ERRα Facilitates the Membrane Fluidity of EC Cells due to the Effects of HMGCS1 on EMT Signaling

Although the role of ERRα in facilitating cancer invasion and metastasis has been reported [28], the underlying mechanisms in EC are not clear. We focused our attention on the role of HMGCS1 in ERRα-mediated invasion and metastasis in EC. First, we identified differentially expressed proteins (DEPs) closely related to ERRα in untreated EC cells and EC cells treated with XCT790 (EC^XCT790^ cells) using tandem-mass-tag-based quantitative proteomics. The following criteria were used: fold change (FC) > 1.2 and *p*-value < 0.05 for upregulated proteins, and FC < 0.83 and *p*-value < 0.05 for down-regulated proteins. All expression values of the DEPs were log transformed and visualized as clustering heatmap (Figure 4a). The quantification results of the DEPs were visualized as volcano plots, as shown in Figure 4b. Fourteen proteins were differentially expressed upon the down-regulation ERRα. Of these proteins, nine were significantly down-regulated, while the other five proteins were up-regulated. According to Gene Ontology analysis, the molecular function classes of the DEPs included hydroxymethylglutaryl-CoA synthase activity, phosphatidylglycerophosphatase activity, acetate-CoA ligase activity, uridine kinase activity, and transferase activity (Figure 4c). Co-immunoprecipitation assays with the indicated antibodies were then performed to confirm the interaction between ERRα and HMGCS1 (Figure 4d). Importantly, a positive relationship was found between these two proteins. HMGCS1 levels increased upon ERRα up-regulation and decreased upon ERRα down-regulation, compared to its levels in the control group (*p* < 0.05, Figure 4e,f). The above-mentioned findings indicate that ERRα and HMGCS1 interact with each other, and that the expression of HMGCS1 is regulated by ERRα. 

Next, the loss-of-function and gain-of-function of ERRα were investigated using a lentiviral infection strategy. To determine whether ERRα promotes cell membrane fluidity and subsequently rearranges the micro-structure of the membrane in an HMGCS1-dependent manner, the expression levels of a series of proteins that are closely related to the cytoskeleton structure of the cell membrane were determined by Western blotting assays. The levels of both matrix metalloproteinase 2 (MMP2) and cortactin were significantly increased in cells overexpressing ERRα and HMGCS1 and decreased in cells with down-regulated expression of ERRα and HMGCS1 (*p* < 0.05, Figure 4g,h). In addition, to further explore whether ERRα and HMGCS1 inhibit EC invasion and metastasis through the EMT signaling pathway, we also determined the expression levels of E-cad and vimentin. The expression level of E-cad was up-regulated and the level of vimentin was down-regulated with the down-regulation of ERRα and HMGCS1. Meanwhile, E-cad was down-regulated and vimentin was up-regulated with the up-regulation of ERRα and HMGCS1. These results indicated that ERRα interacted with HMGCS1 and enhanced cell membrane fluidity via the EMT pathway.

### 2.5. ERRα Promotes Intracellular Cholesterol Metabolism and Enhances the Formation of Invadopodia by Targeting HMGCS1

To confirm our conclusions, we conducted the following series of studies. The loss-of-function and gain-of-function of ERRα were investigated using a lentiviral infection strategy. The expression of HMGCS1 was inhibited by simvastatin. After simvastatin treatment, the expression of HMGCS1 was inhibited in KLE^ERRα-OE^ cells and HEC-1A^ERRα-OE^ cells. Of note, the migration ability of KLE and HEC-1A cells markedly changed after the overexpression of ERRα. Compared to control cells, the scratched spaces increased by 142.1% in KLE^ERRα-OE^ and 64.4% in HCE-1A^ERRα-OE^ cells at 24 h. Meanwhile, the scratched spaces decreased by 36.0% in KLE^ERRα-KD^ cells and by 67.5% in HCE-1A^ERRα-KD^ cells at 24 h compared to the wound in control cells (*p* < 0.05, Figure 5a,b). In addition, the scratched spaces of KLE^ERRα-OE^ and HEC-1A^ERRα-OE^ cells treated with simvastatin for 24 h showed no statistical differences compared to the scratched spaces of the corresponding control cells. Meanwhile, as shown in transwell assays, compared to the control cells, the invasive ability of KLE cells increased approximately two-fold upon ERRα over-expression and reduced by half upon ERRα knockdown. HCE-1A cells showed the same results as KLE cells (*p* < 0.05, Figure 5c,d). Furthermore, the enhanced invasive ability of KLE^ERRα-OE^ and HEC-1A^ERRα-OE^ cells was partially inhibited after simvastatin treatment for 24 h (*p* < 0.05, Figure 5c,d). All of the above-mentioned results indicated that ERRα up-regulated EC cell invasion and metastasis in an HMGCS1-dependent manner.

We found that there were more pseudopodia in ERRα- and HMGCS1-over-expressing EC cells than control EC cells, whereas fewer pseudopodia were seen in cells with down-regulated levels of ERRα and HMGCS1 or those treated with simvastatin, as determined by scanning electron microscopy. Moreover, ERRα over-expression prevented the repression of pseudopod formation by simvastatin. Compared to control cell pseudopodia, ERRα and HMGCS1 promoted the formation of invadopodia and the inhibitive effect of simvastatin on this process was limited when ERRα was up-regulated (Figure 5e). Thus, ERRα increased cell membrane fluidity via the up-regulation of HMGCS1 in KLE and HEC-1A cells.

Cholesterol is essential in maintaining the integrity and fluidity of animal cell membranes [29,30]. Alterations in cholesterol metabolism affect the formation and function of cell membranes, especially lipid rafts and invadopodia, consequently affecting the invasion and metastasis of tumor cells. Previous results of this study confirmed that the up-regulation of ERRα resulted in an increase in HMGCS1 levels, while the down-regulation of ERRα resulted in a decrease in HMGCS1 levels. Therefore, to investigate the role of the ERRα–HMGCS1 axis in cholesterol biosynthesis in EC, a cholesterol assay kit was used. The results showed that, compared to the intracellular cholesterol levels in control EC cells, the intracellular cholesterol levels in ERRα-overexpressing cells increased by almost 33%. Moreover, after ERRα was down-regulated, the levels of intracellular cholesterol decreased by 50%, with similar trends observed in both KLE and HEC-1A cells (*p* < 0.05, Figure 5f,g). However, there were no differences in the levels of extracellular cholesterol between control, ERRα-overexpressing, and ERRα-down-regulated EC cells (*p* > 0.05, Figure 5f,g). 

It is well known that, among the sterol-responsive factors, HMGCS1 is a vital enzyme in the mevalonate pathway. This pathway is the primary regulator of cholesterol synthesis. Therefore, we hypothesized that ERRα facilitates cholesterol metabolism and enhances the formation of invadopodia by targeting HMGCS1.

### 2.6. XCT790 Has an Inhibitory Effect on EC In Vivo

Advanced EC is more likely to invade and metastasize than early-stage EC. Recent studies have shown that ERRα is a potential target for the treatment of EC patients [31]. In our previous study, we showed that XCT790, a specific inhibitor of ERRα, is novel endocrine therapeutic strategy for EC in vitro [32]. The aim of this strategy is to inhibit the expression of ERRα by only using XCT790. Targeting ERRα will concurrently reduce the expression level of HMGCS1, as HMGCS1 is regulated by ERRα.

To further understand the anti-tumor efficacy of XCT790, we performed animal studies using BALB/c nude mice. The mice were randomly divided into two groups (*n* = 5) for intraperitoneal injection (200 µL, 4 mg/kg) of either vehicle or XCT790 every 3 days. Tumor volumes and mouse weights were measured every 3 days, and after drug treatment for 18 days. All mice were euthanized and their tumors were collected. After six consecutive injections, we noted that, compared to the control treatment, XCT790 inhibited tumor growth by 52.66% without a reduction in body weight (Figure 6a,b). The expression of ERRα and HMGCS1 in the tumor tissues of the control and XCT790-treatment groups were analyzed by Western blotting and IHC staining. Both results showed that the expression of ERRα and HMGCS1 were significantly suppressed in the XCT790-treated group compared with the control group (Figure 6c–h). These findings suggested that XCT790 inhibited the expression of ERRα and HMGCS1 and EC growth in vivo. Therefore, XCT790 may be an option for the clinical treatment of EC.

## 3. Discussion

Dyslipidemia is one of the most significant metabolic changes in EC. A greater understanding of the regulatory mechanism of lipid metabolism in EC will facilitate the development of improved therapeutic strategies for EC [33]. We found a biomarker of EC progression based on lipid metabolism, which explains the clinical phenomenon that patients with EC often have abnormal lipid metabolism. Thus, the progression of EC may be inhibited by targeting ERRα. This is also why XCT790 was chosen instead of simvastatin for subsequent animal experiments. Currently, more than half of EC cases can be attributed to obesity, which is considered to be an independent risk factor for this disease [34]. Paying attention to cholesterol metabolism, carrying out appropriate clinical management, and implementing reasonable interventions may reduce the risk of EC and improve the prognosis of patients with EC. 

It is clear that ERRα is a key factor in energy metabolism, but the potential mechanism whereby ERRα regulates the development of EC remains unclear [35]. However, in some hormone-associated tumors, the cholesterol-induced pathway that regulates breast cancer cell metabolism is mediated by ERRα [36], and in prostate cancer, the malignant progression of the disease is caused by the promotion of cholesterol anabolism by ERRα [37]. 

ERRα has potential effects on gynecological diseases. In this study, ERRα was assessed as a therapeutic target for EC. In addition, ERRα may be a novel biomarker for predicting the progression of ovarian endometriosis [38]. Furthermore, as the mechanism of the ERRα-mediated regulation of energy metabolism in EC cells has begun to be revealed, more studies are needed to fully clarify this mechanism [39]. 

ERRα has been confirmed as a vital regulator of metabolism in tumors, participating in glucose metabolism, lipid metabolism, and the tricarboxylic acid cycle, which affect the biological behavior of tumor cells [12,13]. Numerous studies have reported that the upregulation or mutation of HMGCS1 stimulates tumor progression by increasing the in vivo tumor growth and lung metastasis of gastric cancer cells [40] and increasing the cancer stem cell fraction and function in breast cancer [41]. Furthermore, in the tumor microenvironment, the reprogramming of cholesterol metabolism in tumor cells consequently promotes tumor cell initiation and progression [42]. Therefore, we analyzed the clinical data of EC patients using a TMA. The results showed that ERRα and HMGCS1 levels were related to poor prognosis in EC. This is reflected in the MI of EC. Moreover, ERRα and HMGCS1 levels showed a relatively strong correlation in the EC TMA. However, the mechanism of the interaction between ERRα and HMGCS1 remains unclear. 

The interaction between ERRα and HMGCS1 was analyzed by proteomics and co-immunoprecipitation assays, and the results showed a positive correlation between HMGCS1 and ERRα levels. Our previous studies suggested that a high level of ERRα is related to the metastasis and invasion of EC [16]. ERRα and HMGCS1 are considered as predictors of MI due to their relationship to MI in this study. Moreover, we found that higher serum LDL levels were related to LNM in patients with advanced EC. However, with the up-regulation of ERRα expression, intracellular cholesterol levels also increased in EC cells. Therefore, ERRα and HMGCS1 promote lipid metabolism in EC and are involved in the invasion and metastasis of EC. The relationship between ERRα and EMT in EC cells showed that the down-regulation of ERRα inhibited transforming-growth-factor-β-induced EMT [43]. Essentially, our study also showed that the regulation of ERRα and HMGCS1 affected the expression of the vital EMT pathway proteins, E-cad and vimentin. This is the first report of the interaction between ERRα and HMGCS1 and its involvement in invadopodia formation through the EMT signaling pathway in EC.

Invadopodia play a key role in metastatic tumor cells [44], as they can lead to extracellular matrix degradation, local migration and invasion, and transmission to a distant organ via blood vessels [25]. There are 129 invadopodia-associated proteins belonging to different functional classes, including matrix metalloproteases (MMPs), cortactin, and Src family tyrosine kinases [45]. MMPs, especially MMP2, promote proteolytic matrix degradation and contribute to EMT progression to favor cancer cell invasion [46]. In addition, cortactin, a substrate of Src family tyrosine kinases, is responsible for invadopodia assembly by regulating the F-actin-enriched invadopodia cytoskeleton to promote cancer migration, invasion, vessel penetration, and metastasis [47,48]. Invadopodia are a type of cell membrane structure rich in cholesterol. When invadopodia form, several invadopodia proteins, such as cortactin, receptors, signaling adaptors, and trafficking proteins are assembled. Hence, if cholesterol synthesis is impaired, cell membrane fluidity and invadopodia formation may be inhibited [49]. We chose to analyze cortactin, MMP2, and cholesterol to evaluate the formation and the function of invadopodia in this study. Alterations in the levels of ERRα, HMGCS1, and cholesterol influence lipid raft construction, thus affecting the formation of invadopodia and invadopodia-mediated invasion and metastasis of EC. However, it is still unclear how invadopodia are regulated physiologically. Of note, previous studies have revealed that metformin decreases the rate of lipolysis and lipid peroxidation in transgenic mouse models of EC, thereby inhibiting EC metastasis [9]. These results may provide new perspectives on the role of lipid metabolism in tumor metastasis. 

Invadopodia are known as a phosphatidylinositol- and cholesterol-enriched microdomains on cell membranes [50]. Our previous study found that the overexpression of ERRα was associated with EC invasion and metastasis, although the mechanism remained unclear [16]. In the present study, ERRα was found to significantly increase the intracellular levels of cholesterol in EC, resulting in clearly enhanced cell membrane fluidity, together with an increase in MMP2 and cortactin levels. Therefore, it was confirmed that lipid metabolism is involved in the formation of pseudopodia and facilitates the metastasis of tumor cells [51,52]. Moreover, we reported a novel function of HMGCS1 during EC invasion. In this study, we confirmed that the expression of HMGCS1 is regulated by ERRα. Therefore, the over-expression or knockdown of ERRα resulted in an increase or decrease, respectively, in HMGCS1 levels. The levels of E-cad and vimentin also decreased with ERRα and HMGCS1 down-regulation. In addition, the down-regulation of ERRα and HMGCS1 markedly reduced cholesterol levels and membrane fluidity and decreased the levels of cortactin and MMP2 and their function in the invasive progression of EC. Similar results were also observed in EC cells treated with simvastatin. In addition, E-cad and vimentin, the star proteins of the EMT signaling pathway, were also down-regulated and indicated that the ERRα–HMGCS1 axis regulates EC through EMT progression. Our findings verified that ERRα positively regulates the migration and invasion of EC via HMGCS1-mediated intracellular cholesterol reprogramming, enhanced cell membrane fluidity, and the promotion of invadopodia formation and EMT progression. 

Of course, there are some limitations of this study that should be noted. Prognostic follow-up data for the enrolled patients are lacking, and it is necessary to validate our findings in larger cohorts or randomized controlled trials in the future. In addition, simvastatin does not specifically down-regulate the expression of HMGCS1. Moreover, organoid experiments should be performed to establish a more realistic and effective disease model. Finally, molecular typing was not mentioned, although estrogen-receptor-positive and -negative cell lines were selected for our experiments.

## 4. Materials and Methods

### 4.1. Cell Lines and Cell Culture

KLE cells (KeyGEN BioTECH, Nanjing, China) were cultured in RPMI 1640 (#R8758, Sigma-Aldrich, St. Louis, MO, USA) and HEC-1A cells (KeyGEN BioTECH, Nanjing, China) were maintained in McCoy’s 5 A medium (#M9309, Sigma-Aldrich, St. Louis, MO, USA), supplemented with 10% FBS (#10091148, Gibco, Billings, MT, USA) and 1% antibiotic-antimycotic solution (#B120901, BasalMedia, Shanghai, China). All these cells were routinely maintained in a humidified incubator at 5% CO_2_ at 37 °C. Cells treated with Simvastatin (#79902-63-9 Sigma-Aldrich, St. Louis, MO, USA) were cultured in phenol red-free medium (#21041025, Thermo Fisher, Waltham, MA, USA) containing 1% Serum Replacement 1 (#S0638 Sigma-Aldrich, St. Louis, MO, USA). KLE and HEC-1A cells were, respectively, cultured with 10 μM and 5 μM Simvastatin (in dimethyl sulfoxide [DMSO]; #Y190601, MP Biomedicals LLC, Santa Ana, CA, USA) or DMSO (control) for 24 h. Then, the lentiviral vectors were used to over-express ERRα and achieved in KLE and HEC-1A cells and named KLE^ERRα-OE^ and HEC-1A^ERRα-OE^, respectively. Lentiviral vectors expressing siRNAs targeting ERRα (labelled ERRα-KD) were established and transfected into KLE and HEC-1A cells. The target sequences of si-ERRα (GenBank accession NM_004451.5) was 5′-GAG CGA GAG TAT GTT CTA-3′. In addition, KLE and HEC-1A cells were named KLE^ERRα-KD^ and HEC-1A^ERRα-KD^ with ERRα down-expression by lentivirus-mediated siRNA, respectively.

### 4.2. Immunohistochemistry (IHC) on Patients Tissues

All patients’ tissues immunostaining for *HMGCS1* and *ERRα* was performed based on standard procedures. Rabbit polyclonal anti-*HMGCS1* (dilution 1:100; #ab155787, Abcam, London, UK) and rabbit polyclonal anti-*ERRα* (dilution 1:100; #ab137489, Abcam, London, UK) antibodies were employed. The percentage of positive cells was scored as 0 (cells < 5 percent), 1 (5 to 25 percent), 2 (26 to 50 percent), 3 (51 to 75 percent), and 4 (76 to 100 percent). Positive staining intensity was scored as 0 (no positive), 1 (weak positive), 2 (moderate positive), and 3 (strong positive). The expression of *HMGCS1* and *ERRα* was measured by the immunoreactive scores (IRSs) using the algorithm: IRS = Si × Pi (where Si and Pi represent the intensity and percentage of cells with positive staining, respectively). Samples were split into four groups according to their IRS: 0, negative (−); 1–4, weakly positive (+); 5–8, positive (++); 9–12, strongly positive (+++). *ERRα* and *HMGCS1* expression of mice tumor tissue were evaluated using the fully automated VIS DIA VisioMorph system (Visiopharm^®^, Hoersholm, Denmark), using similar image processing principles as described previously. In brief, all TMA-slides of mice were scanned at 40× magnification using a Leica SCN400 slide scanner (Leica Biosystems, Wetzlar, Germany) and imported into the image analysis software program Visiopharm^®^, and a digital image was recorded of each core. The expression levels of *HMGCS1* and *ERRα* were measured by the Histochemistry score (H-Score) using the algorithm: H-Score = ∑(pi × i) = (percentage of weak intensity × 1) + (percentage of moderate intensity × 2) + (percentage of strong intensity × 3), where pi represents the pixel area/cell number ratio of positive signal and i stands for tint intensity. H-score was between 0 and 300. The larger the H-score is, the stronger the comprehensive positive intensity is.

### 4.3. Western Blotting (WB)

The cultured cell lysates were obtained by RIPA lysis buffer (#P0013B, Beyotime Biotechnology, Shanghai, China) with protease inhibitors (#87786, Thermo Fisher, Waltham, MA, USA). Then, protein was extracted, quantified, separated by SDS-PAGE and transferred onto polyvinylidene fluoride membranes. After being blocked with bovine serum albumin, the target protein was probed with antibodies against human *E-cadherin* (1:1000; #3195, Cell Signaling Technology), *ERRα* (1:500; #ab137489, Abcam, MA, USA), *HMGCS1*(1:1000; #17643-1-AP, Proteintech, Wuhan, China), *Cortactin* (1:4000; #11381-1-AP, Proteintech, Wuhan, China), *GAPDH* (1:3000; #60004-1-Ig, Proteintech, Wuhan, China), MMP2 (1:2000; #10372-2-AP, Proteintech, Wuhan, China), *Vimentin* (1:1000; #5741, Cell Signaling Technology), next incubated with secondary antibodies (1:6000; 66031-1-Ig, Proteintech, Wuhan, China), and detected with enhanced chemiluminescence reagents (#34577, Thermo Fisher Scientific, Waltham, MA, USA). Finally, protein bands were imaged by FluorChem M automatic chemiluminescence image analysis system (Protein Simple, San Jose, CA, USA).

### 4.4. Wound Healing Assay

EC cells were seeded to confluence in 6-well plates and 200-μL tips were employed to introduce a scratch in the monolayer. Scratch widths were imaged at 0 and 24 h. Changes in migration area were observed using ImageJ software (NIH, Bethesda, MA, USA). The horizontal migration rate was calculated according to the following formula: (width 0 h − width 24 h)/width 0 h × 100%.

### 4.5. Transwell Chamber Invasion Assay

Matrigel™ Basement Membrane Matrix (40 μL; #354234, BD, USA) was added to 24-well (8 μm) plates containing transwell chambers (#3422, Corning, NY, USA) to form Matrigel-coated membrane for the subsequent experiment. Briefly, 200 μL of cell suspension (2.5 × 10^5^ cells/mL) with 1% FBS was added to the upper chambers and 1300 μL of medium containing 10% FBS was supplemented to the lower chamber. After incubation at 37 °C for 24 h, remove the non-invading cells of the upper chambers with a cotton swab, and the filters were fixed with paraformaldehyde and stained with crystalline violet. The stained cells were imaged at X200 magnification using an inverted microscope (Olympus, Tokyo, Japan).

### 4.6. Tandem Mass Tag (TMT) Labeling Proteomics

The total protein of EC cells and EC^XCT790^ cells dealt with 10μM XCT790 were extracted and evaluated by sodium dodecyl sulfate-polyacrylamide gel electrophoresis (SDS-PAGE) and staining, which included ECC-1, HEC-1, KLE, ECC-1^XCT790^, HEC-1^XCT790^ and KLE^XCT790^ cells. In brief, cells were lysed firstly. Sequence-grade modifiedtrypsin (Promega, Madison, WI, USA) was used to digest the proteins and the resultant peptide mixture was labeled based on the TMT kit (ThermoFisher Scientific, USA). After desalination, high-pH HPLC was used for grading. Then, protein fractions were analyzed by online electrospray tandem mass spectrometry. Differentially expressed proteins were selected according to fold change and *p* value.

### 4.7. Co-Immunoprecipitation Assay (Co-IP)

The whole cell lysates were extracted with RIPA lysis buffer (#P0013B, Beyotime Biotechnology, Shanghai, China) accompanied by protease inhibitors (#87786, Thermo Fisher, Waltham, MA, USA) on ice according to the instruction. For Co-IP, 500 µg protein was incubated with 1–2 µg specific antibodies at 4 °C overnight with constant rotation; 30 µL of 50% Protein A/G PLUS-Agarose beads (#sc-2003, Santa Cruz Biotechnology, Santa Cruz, USA) was then added, and the incubation proceeded for another four hours at 4 °C. Beads were then washed four times with the lysis buffer. Between washes, the beads were collected by centrifugation at 10,000 g for 30 sec at 4 °C. After a final wash, the supernatant was aspirated and discarded, the precipitated proteins were eluted from the beads by resuspending twice in SDS-PAGE loading buffer and boiling for 5 min. The resultant materials underwent Western blot analysis.

### 4.8. Scanning Electron Microscope (SEM)

KLE and HEC-1A cells were seeded in 24-well plates containing glass coverslips with electron microscopy fixative (#G1102, Servicebio, Wuhan, China). After a post-fixing overnight at 4 °C, the cells were dehydrated in a series of graduated ethanol ranging from 30% to 100% (#100,092,183 Sinaopharm Group Chemical Reagent Co. LTD, Shanghai, China) for 15 min, respectively. Use Critical Point Dryer to dry samples. Then, the specimens were coated with gold by the SCD 500 sputter coater for 30 s. Additionally, the cellular morphology was observed by using a HITACHI-SU8100 scanning electron microscope.

### 4.9. Cholesterol Quantitation Assay

The level of cholesterol was quantitated by using the esterol/Cholesteryl Ester Quantitation Assay kit (ab102515, Abcam, MA, USA). Briefly, EC cells with indicated treatment were centrifuged and the deposit was re-suspended in the mixture of Chloroform: Isopropanol: NP-40 (7:11:0.1) solution using a micro-homogenizer. Then, the liquid in the organic phase was obtained after being spun, dried, and vacuumed. Additionally, then, the concentration of cholesterol was quantified according to the manufacturer’s protocol. The OD value of each sample was measured by spectrophotometer (λ = 570 nm).

### 4.10. Participants and Specimens

The blood samples and tissue samples of the EC group and normal endometrial group were gained from the patients who underwent surgical therapy at Fujian Provincial Maternity and Child Health Hospital, Affiliated Hospital of Fujian Medical University from 2012 to 2021. None of the patients received any chemotherapy, radiation, or hormonal therapy before surgery. Among the specimens, we collected 161 blood samples, including 129 EC patients and 32 normal people in Figure 1; and we collected 103 tissue samples, including 73 EC patients and 30 normal people in Figure 2. Furthermore, the diagnoses of the pathological sections which were embedded in paraffin were made by skilled professional pathologists. Informed consent was received from all patients. This research protocol received support and approval from the Ethics Committee of Fujian Maternity and Child Health Hospital affiliated with Fujian Medical University (No. FMCH-2021-KRD030).

### 4.11. Animal Studies

KLE cells (9 × 10^6^ cells in 0.2 mL of serum-free DMEM/injection) were subcutaneously injected into the upper flank region of BALB/c nude female mice (about five weeks old). Treatment was initiated when tumor volumes reached a mean size of approximately 100 mm^3^. One month after cell inoculation, the mice, divided into two groups (*n* = 5) were subsequently subjected to intraperitoneal injections of saline or XCT790 (4 mg/kg) every three days for 18 days, a total of six times. Tumor volumes were measured every three days with calipers and calculated by using the following formula: volume (mm^3^) = 0.5 × (length) × (width)^2^. After being treated for 18 days, mice were euthanized, and tumors were resected and immersed in 4% paraformaldehyde. The effects of XCT790 on *ERRα* and *HMGCS1* were analyzed by Western blotting (WB) and immunohistochemistry (IHC) staining. The study was approved by the Committee for Animal Experiments of Fujian Maternity and Child Health Hospital affiliated with Fujian Medical University (No. FMCH-2021-KRD030).

### 4.12. Statistical Analysis

All data were analyzed by IBM SPSS (version 22) and GraphPad Prism 8.0 software. Statistical analysis was operated with at least three separate experiments under the same conditions. Quantitative variables were expressed as mean ± SD and analyzed by Student’s *t*-test or one-way ANOVA. Qualitative variables were then compared using Pearson’s χ^2^ test or Fisher’s exact test. In addition, related parameters were analyzed by using Pearson’s rank correlation analysis. R/Bioconductor software was used for all bioinformatics analysis. Statistical significance was set at *p*-values less than 0.05 (*, *p* < 0.05).

## 5. Conclusions

In summary, our study revealed that the ERRα–HMGCS1 axis promoted the formation of invadopodia, matrix degradation, EMT, and the metastasis of EC. This regulation was dependent on the participation of HMGCS1 in lipid reprogramming and cellular membrane remodeling. An increase in ERRα levels in EC results in HMGCS1 upregulation to enhance cholesterol biosynthesis and promote cellular fluidity, consequently affecting EC invasion and metastasis via lipid rafts and invadopodia. So, we provide new evidence of ERRα/HMGCS1 as a potential biomarker of poor prognosis in EC. The issue explains the role of ERRα/HMGCS1 in EC and demonstrates a theoretical foundation for metastatic EC treatment based on ERRα/HMGCS1.

## Figures and Tables

**Figure 1 ijms-24-04010-f001:**
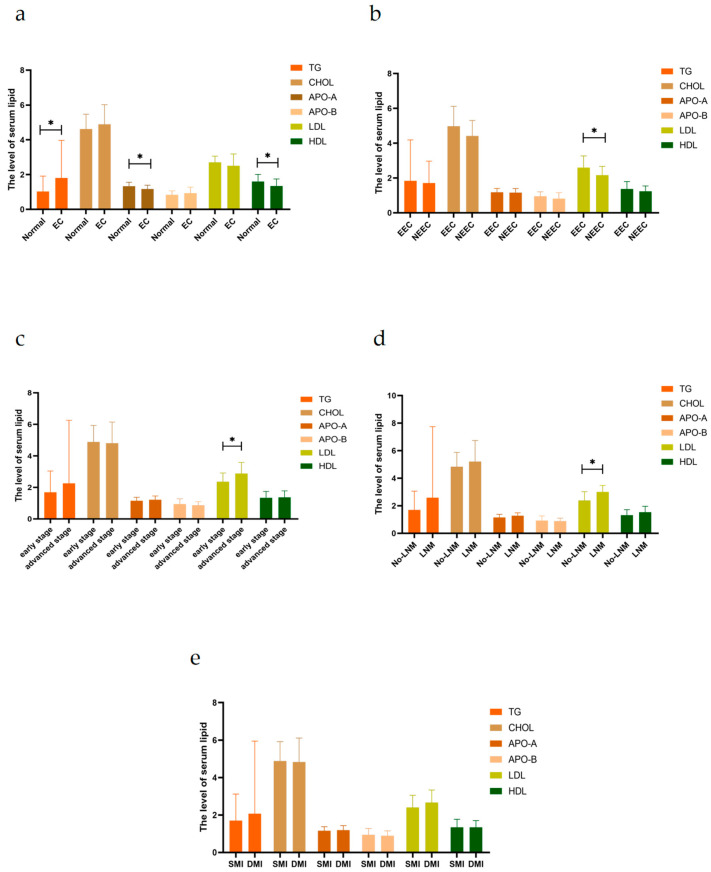
Serum lipid levels in patients with EC are associated with the progression of EC. (**a**) Compared to the healthy population, endometrial cancer (EC) patients had higher triglyceride (TG) levels and lower apolipoprotein A (APO-A) and high-density lipoprotein (HDL) levels. (**b**) Patients with endometrioid endometrial cancer(EEC) had higher low-density lipoprotein(LDL) levels than patients with non-endometrioid endometrial cancer(NEEC). (**c**) Patients with advanced-stage EC had higher LDL levels than those with early-stage EC. (**d**) Patients with lymph node metastasis (LNM) had higher LDL levels than those without LNM. (**e**) Patients with deep myometrial invasion (≥1/2 MI) had higher TG levels and lower LDL levels than those with shallow myometrial invasion MI (<1/2 MI), but there was no statistically significant difference.Abbreviations: TG Triglyceride; CHOL Total Cholesterol; Apo-A Apolipoprotein A; Apo-B Apolipoprotein B; LDL Low-Density Lipoprotein; HDL High-Density Lipoprotein; ECC Endometrioid Endometrial Cancer; NEEC Non-Endometrioid Endometrial Cancer; LNM Lymph Node Metastasis; MI Myometrial Invasion; SMI (<1/2 MI) Shallow Myometrial Invasion; DMI (≥1/2 MI) Deep Myometrial Invasion. *, indicates statistical significance. Statistical tests: Student’s *t*-test.

**Figure 2 ijms-24-04010-f002:**
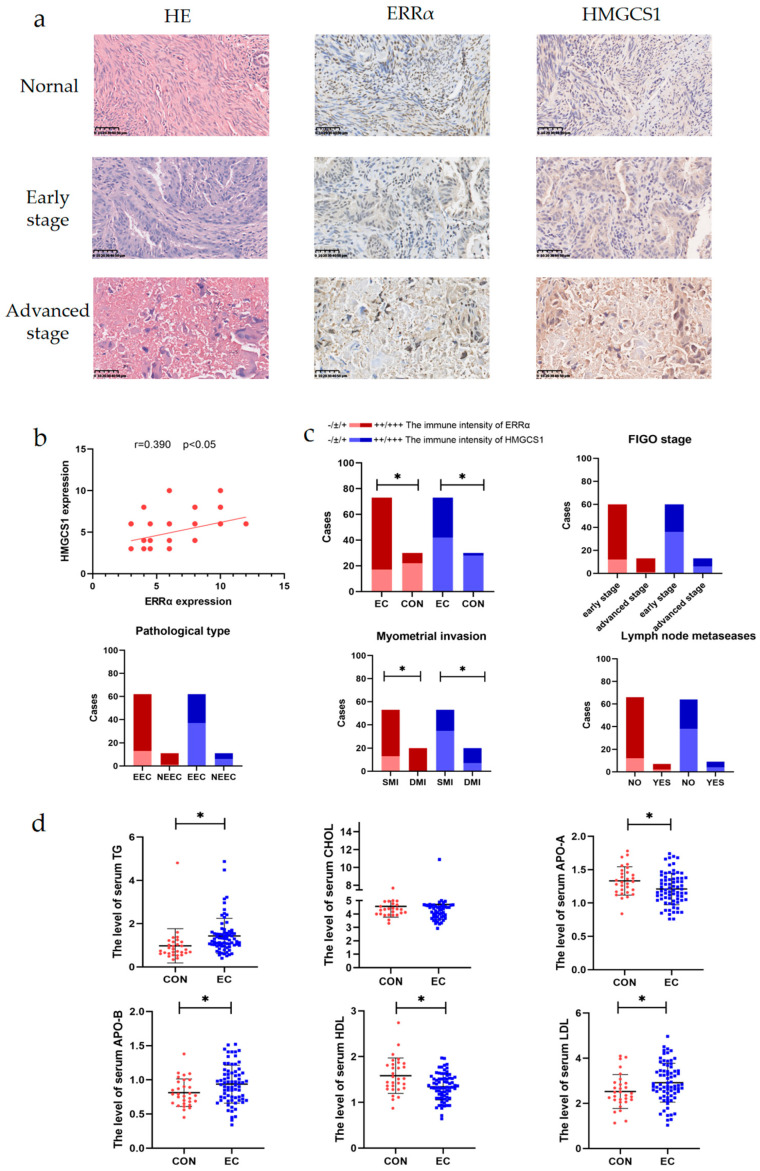
ERRα and HMGCS1 are highly expressed in the tissues of patients with EC and are associated with metastasis. (**a**) The expression of ERRα and HMGCS1 in EC tissues at different International Federation of Gynecology and Obstetrics (FIGO) stages and control endometrial tissues (magnification, 400×; scale bars, 50 μm). (**b**) the correlation analysis of between ERRα and HMGCS1 expression levels of EC patients. (**c**) Analysis of ERRα and HMGCS1 and histopathological characteristics. (**d**) Differences in the serum levels of TG, cholesterol (CHOL), APO-A, APO-B, HDL, and LDL in healthy control patients and patients with EC. *, indicates statistical significance. Statistical tests: Student’s t-test and Pearson’s rank correlation analysis.

**Figure 3 ijms-24-04010-f003:**
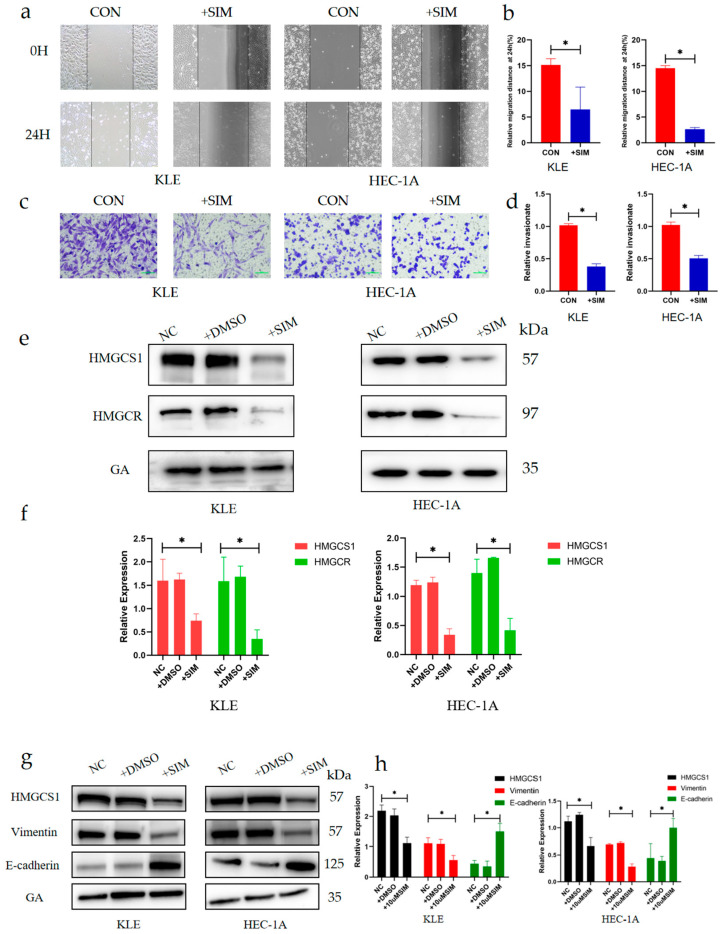
Simvastatin inhibits the epithelial-mesenchymal transition in EC. (**a**–**d**) The migration abilities of EC cells after simvastatin treatment and statistical analysis results (magnification, 200×; scale bars, 100 μm). (**e**,**f**) The expression levels of 3-hydroxy-3-methyl-glutaryl-coenzyme A reductase (HMGCR) and HMGCS1 after up- and down-regulation of ERRα expression, and the statistical analysis results. (**g**,**h**) The expression levels of EMT-related proteins after simvastatin treatment. SIM: Simvastatin. *, indicates statistical significance. Statistical tests: Student’s *t*-test.

**Figure 4 ijms-24-04010-f004:**
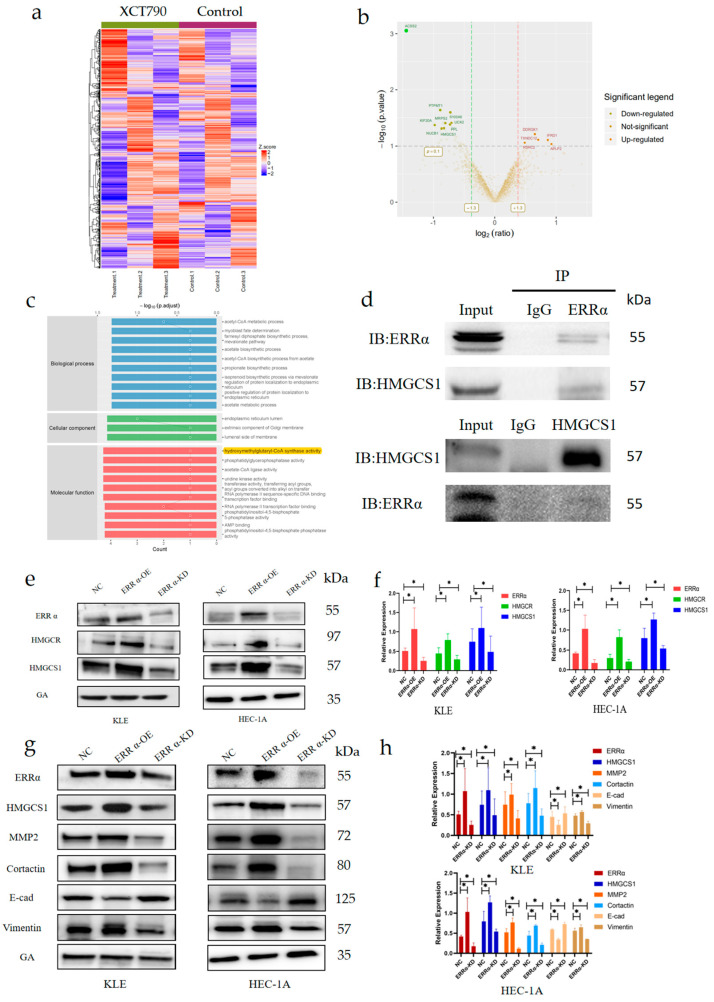
ERRα facilitates the membrane fluidity of EC cells due to the effects of HMGCS1 on EMT signaling. (**a**) After comparing untreated EC cells and EC cells treated with 10 μM XCT790 for 24 h (EC^XCT790^) by proteomics, proteins related to ERRα were identified by hierarchical cluster analysis. (**b**) Volcano plots of the 14 proteins significantly correlated with ERRα. The x-axis is the log 2-fold change (FC), and the y-axis is the −log 10 (*p*-value). (**c**) Gene Ontology annotation classification analysis of proteins associated with ERRα down-regulation, the proteins were classified according to biological process (BP), cellular component (CC), and molecular function (MF). (**d**) Co-immunoprecipitation verified that ERRα and HMGCS1 interact with each other. (**e**,**f**) The expression levels of HMGCR and HMGCS1 after ERRα up- and down-regulated in EC cells and the statistical analysis results. (**g**,**h**) The effect of ERRα and HMGCS1 on the expression levels of EMT-related and membrane fluidity biomarkers, E-cadherin (E-cad), vimentin, matrix metalloproteinase 2 (MMP2), and cortactin after ERRα and HMGCS1 up- and down-regulation in EC cells, and the statistical analysis results. *, indicates statistical significance. Statistical tests: Student’s *t*-test or ANOVA.

**Figure 5 ijms-24-04010-f005:**
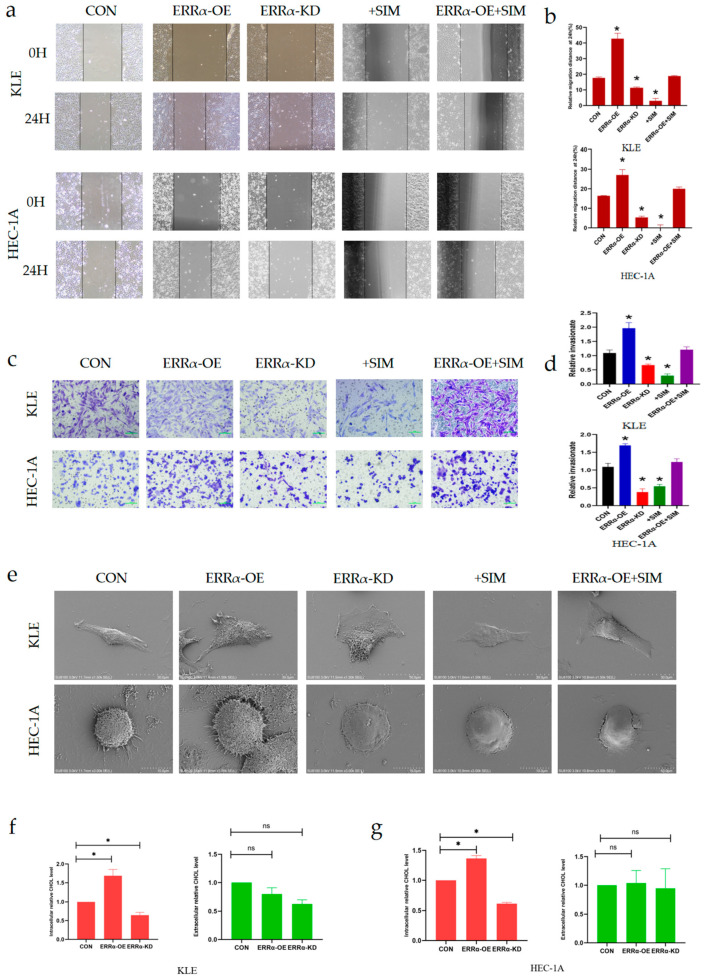
ERRα promotes intracellular cholesterol metabolism and enhances the formation of invadopodia by targeting HMGCS1**.** (**a**,**b**) Effects of the ERRα–HMGCS1 axis and simvastatin treatment for 24 h on wound healing in ERRα-over-expressing and control EC cells, and the corresponding statistical analysis results (magnification, 200×; scale bars, 100 μm). (**c**,**d**) Effects of the ERRα–HMGCS1 axis and simvastatin treatment for 24 h on transwell chamber invasion by ERRα-over-expressing and control EC cells, and the corresponding statistical analysis results (magnification, 200×; scale bars, 100 μm). (**e**) Representative scanning electron microscope (SEM) micrographs of KLE and HEC-1A cells. Pseudopod formation was affected by the ERRα–HMGCS1 axis and simvastatin treatment for 24 h (scale bar, 100 μm). (**f**,**g**) The intracellular and extracellular levels of cholesterol in EC cells after ERRα and HMGCS1 up- and down-regulation, as detected using a colorimetric method. *, indicates statistical significance. Statistical tests: Student’s *t*-test or ANOVA.

**Figure 6 ijms-24-04010-f006:**
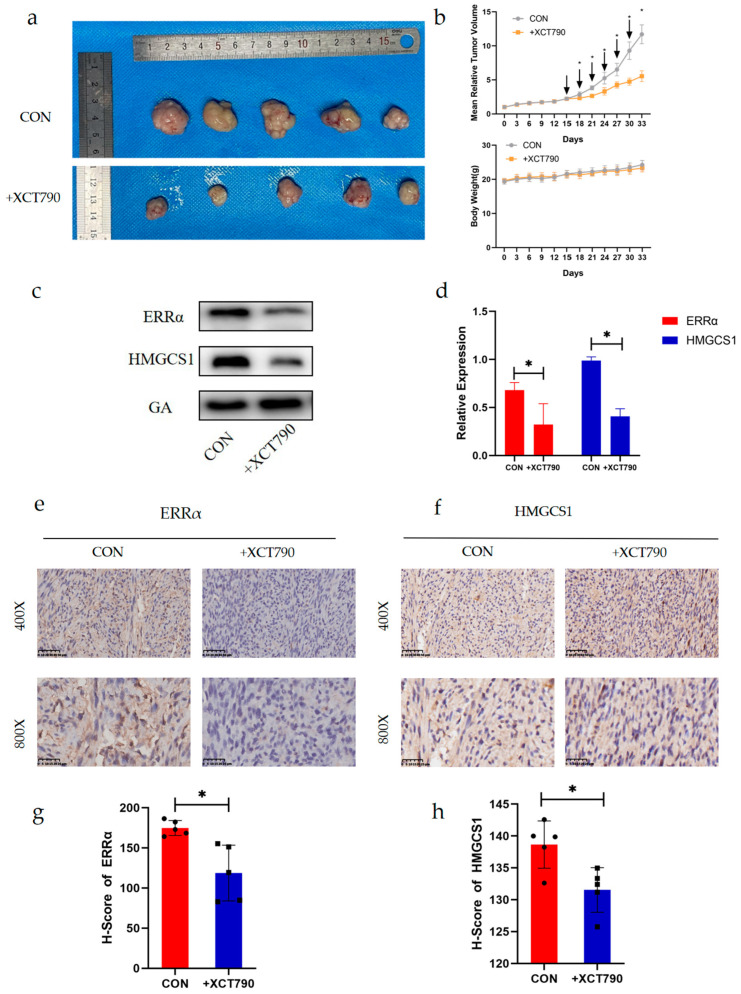
The in vivo anti-cancer efficacy of XCT790 in EC. (**a**) Tumor images after 18 days of treatment. (**b**) Changes in tumor volume and body weight. The indicated compounds (4 mg/kg) were intraperitoneally injected every 3 days, for a total of six injections, including vehicle injections (1% dimethyl sulfoxide in 0.9% NaCl) for the control group (*n* = 5). (**c**,**d**) The expression levels of ERRα and HMGCS1 in subcutaneous orthotopic xenograft tissues determined by Western blotting. (**e**–**h**) The expression levels of ERRα and HMGCS1 in subcutaneous orthotopic xenograft tissues, determined by immunohistochemical analysis, and the corresponding statistical analysis results (magnification, 400×; scale bars, 50 μm; magnification, 800×; scale bars, 25 μm). *, indicates statistical significance. Statistical tests: Student’ s *t*-test.

**Table 1 ijms-24-04010-t001:** Histopathological results of patients with EC.

		Number of Cases	Ratio (%)
FIGO Staging	Stage I	47	64.38
	Stage II	13	17.81
	Stage III-IV	13	17.81
Pathological Type	EEC	61	83.56
	I	31	50.82
	II	28	45.90
	III	5	8.20
	NEEC	16	21.92
Myometrial Invasion	<1/2	53	72.60
	≥1/2	20	27.40
Lymph Node Metastasis	NO	66	90.41
	YES	7	9.6
Total		73	100

Ratio (%): the percentage of each subgroup in the total number of cases (73 cases) in EC patients; Abbreviations: EC Endometrial Cancer, EEC Endometrioid Endometrial Cancer, NEEC Non-Endometrioid Endometrial Cancer.

**Table 2 ijms-24-04010-t002:** Association between ERRα, HMGCS1 and ERRα /HMGCS1 expression and clinical variables in a tissue microarray of endometrial cancer and control tissues.

	ERRα Expression			HMGCS1 Expression			ERRα/HMGCS1 Expression		
	−/±/+	++/+++	*p*	−/±/+	++/+++	*p*	Both ++/+++	Either −/±/+	*p*
Nornal	22	8	*p* < 0.001 *	28	2	*p* < 0.001 *	0	30	*p* < 0.001 *
EC	17	56		42	31		26	47	
FIGO staging									
Early stage	12	48	*p* = 0.515	36	24	*p* = 0.360	19	41	*p* = 0.232
Advanced stage	1	12		6	7		7	6	
Pathological Type									
EEC	13	49	*p* = 0.411	37	25	*p* = 1.00	20	42	*p* = 0.280
NEEC	1	10		6	5		6	5	
Myometrial Invasion									
<1/2	13	40	*p* = 0.036 *	35	18	*p* = 0.017 *	13	40	*p* = 0.001 *
≥1/2	0	20		7	13		13	7	
Lymph Node Metastasis									
NO	12	54	*p* = 0.392	38	26	*p* = 0.625	21	45	*p* = 0.096
YES	2	5		4	5		5	2	

Abbreviations: ERRα Estrogen-Related Receptor α, HMGCS1 3-hydroxy-3-methylglutaryl-CoA synthase 1. *p* < 0.05 suggests significant difference. *, indicates statistical significance. Statistical tests: Pearson’s χ^2^ test.

**Table 3 ijms-24-04010-t003:** The serum lipid levels of patients with endometrial cancer and control subjects.

	Controls	EC	*p* Value
TG	0.789 ± 0.144	1.434 ± 0.095	*p* = 0.011 *
CHOL	4.569 ± 0.146	4.715 ± 0.132	*p* = 0.523
APO-A	1.332 ± 0.039	1.211 ± 0.028	*p* = 0.030 *
APO-B	0.813 ± 0.037	0.937 ± 0.033	*p* = 0.001 *
HDL	1.584 ± 0.071	1.346 ± 0.034	*p* = 0.001 *
LDL	2.522 ± 0.137	2.915 ± 0.101	*p* = 0.032 *

Abbreviations: TG Triglyceride, CHOL Cholesterol, APO-A Apolipoprotein A, APO-B Apolipoprotein B, HDL High-Density Lipoprotein, LDL Low-Density Lipoprotein. *p* < 0.05 suggests significantly difference. *, indicates statistical significance. Statistical tests: Student’ s *t*-test.

## Data Availability

The datasets analyzed during the current study are available from the corresponding author on reasonable request.

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
