# Peer review of "ERRα Up-Regulates Invadopodia Formation by Targeting HMGCS1 to Promote Endometrial Cancer Invasion and Metastasis"

_ijms, 2023, doi:10.3390/ijms24044010_

Round 1

Reviewer 1 Report

Dear Editor, thank you so much for inviting me to revise this manuscript about endometrial cancer.

This study addresses a current topic.

The manuscript is quite well written and organized. English should be improved.

Figures and tables are comprehensive and clear.

The introduction explains in a clear and coherent manner the background of this study.

We suggest the following modifications:

·      Introduction section: although the authors correctly included important papers in this setting, we believe a couple of studies should be cited within the introduction ( PMID: 35887962; PMID: 35807197), only for a matter of consistency. We think it might be useful to introduce the topic of this interesting study.

·      Methods and Statistical Analysis: nothing to add.

·      Discussion section: Very interesting and timely discussion. Of note, the authors should expand the Discussion section, including a more personal perspective to reflect on. For example, they could answer the following questions – in order to facilitate the understanding of this complex topic to readers: what potential does this study hold? What are the knowledge gaps and how do researchers tackle them? How do you see this area unfolding in the next 5 years? We think it would be extremely interesting for the readers.

However, we think the authors should be acknowledged for their work. In fact, they correctly addressed an important topic, the methods sound good and their discussion is well balanced.

One additional little flaw: the authors could better explain the limitations of their work, in the last part of the Discussion.

We believe this article is suitable for publication in the journal although major revisions are needed. The main strengths of this paper are that it addresses an interesting and very timely question and provides a clear answer, with some limitations.

We suggest a linguistic revision and the addition of some references for a matter of consistency. Moreover, the authors should better clarify some points.

Reviewer 2 Report

This study aims to investigate the role of ERRa and HMGCS1 in regulating intracellular cholesterol metabolism to promote EC progression. A direct interaction between ERRa and HMGCS1 was evidenced by proteomic and co-immunoprecipitation analysis. Immunohistochemistry illustrated that ERRa and HMGCS1 are associated to EC progression. Loss-of-function and gain-of-function assays revealed that ERRa/HMGCS1 overexpreesion promote intracellular cholesterol metabolism and that their inhibition significantly weaken the malignant progression of EC both in vitro as in vivo analysis. Authors demonstrate that ERRa promotes EC invasion and metastasis through the HMGCS1-mediated intracellular cholesterol metabolism pathway, which is dependent of the EMT pathway, then suggesting that the ERRa/HMGCS1 axis is a potential target to suppress EC progression.

From the experimental point of view the study is well conducted, with appropriate methodologies, and the conclusions are well founded with specific experiments. The study provides a good translational component including a couple of cell lines models, an in vivo approach and also an analysis of specific biomarkers in serum and tissue from a patient cohort.

However, from my view I consider that the manuscript might be improved with the incorporation of additional analysis:

1. A new variable of the analysis including the co-expression of ERRa/HMGCS1 would be very useful. Correlations of this variable with clinical variables are required.

2. There is not any information on the follow-up of the patients included in the analysis. Include the follow-up and the association of the studied variables with the progression free-interval (log-rank tests with Kaplan-Meier plots and multivariate analysis). It would show how the biomarkers behave from the prognostic point of view. 

3. It is not well understood that authors use Simvastatin for the in vitro analysis and XCT790 for the in vivo models. Which are the reasons of these approaches? Why the authors do not use the same chemical for both experiments? Please justify this approach and if it is possible present in vitro evidences of the effects of XCT790.

4. Do the authors think that Simvastatin, XCT790 or metformin might have a role as coadjutant agent in the management of EC patients? Please discuss this possibility in view with the reported results.

Minor comments:

- Please check the names of genes, they must be expressed in Italics.

- Express the 'p' of the p-values of the tables in lowercase.

- Figure 2 and Figure 6. The detail of the histological and immunohistochemical images is very poor and needs to be improved (higher magnification).

- Figure 4. GEO annotation is not clearly visible. It needs to be improved.

Round 2

Reviewer 1 Report

acceptance.

Author Response

Dear Reviewer,

Thank you for your advise and affirmation.

Reviewer 2 Report

I would like to thank the author the responses provided to my revision and also the attempt to clarify and improve the content of the manuscript. However, there are still some concerns that from my view are not completely addressed:

1 (Point 2). About the cohort of patients included there is not any information about the follow-up and on the impact of the analyzed biomarkers with regards the prognosis. The response provided by the authors is not enough and it doesn't justify why these analysis have not been done. Why this data is lacking? This information would support and significantly increase the scientific value of the authors' findings.

2 (Point 3). It is not well understood that authors use Simvastatin for the in vitro analysis and XCT790 for the in vivo models. Which are the reasons of these approaches? Why the authors do not use the same chemical for both experiments? These questions have not been answered.

3 (Point 4). This point 4 is not well discussed... the effects of simvastatin and XCT790 on in vivo models seem similar... I insist that it is not well justified the use of different compound in the in vitro and the in vivo models... 

To be considered for duicussion:

- Kim JS, Turbov J, Rosales R, Thaete LG, Rodriguez GC. Combination simvastatin and metformin synergistically inhibits endometrial cancer cell growth. Gynecol Oncol. 2019 Aug;154(2):432-440. doi: 10.1016/j.ygyno.2019.05.022. Epub 2019 Jun 6. PMID: 31178149.

Kokabu T, Mori T, Matsushima H, Yoriki K, Kataoka H, Tarumi Y, Kitawaki J. Antitumor effect of XCT790, an ERRα inverse agonist, on ERα-negative endometrial cancer cells. Cell Oncol (Dordr). 2019 Apr;42(2):223-235. doi: 10.1007/s13402-019-00423-5. Epub 2019 Jan 31. PMID: 30706380.

Regarding some statements provided in the response to the reviewer... please include the references in which authors support their assertions. For instance: 'Moreover, in the animal experiments, as expected, a decrease in HMGCS1 expression levels were also observed after XCT790 treatment in vivo, as determined by IHC assays. Currently, targeted anticancer drugs combined with simvastatin are the principal therapeutic strategies for EC instead of simvastatin alone (???). Importantly, there are also sufficient anticancer effects when using only XCT790, instead of XCT790 combined with simvastatin, and this has less toxicity to animals (?)'.

Author Response

Dear reviewer,

Thank you for your advise and comments. I have uploaded my responses as a Word file. Please see the attachment.
